# Design and Optimization of Sensitivity Enhancement Package for MEMS-Based Thermal Acoustic Particle Velocity Sensor [note 1]

**DOI:** 10.3390/s21134337

**Published:** 2021-06-24

**Authors:** Wenhan Chang, Lingmeng Yang, Zhezheng Zhu, Zhenchuan Yang, Yilong Hao, Chengchen Gao

**Affiliations:** Institute of Microelectronics, Peking University, Beijing 100871, China; whchang@pku.edu.cn (W.C.); lm_yang16@pku.edu.cn (L.Y.); triplez@pku.edu.cn (Z.Z.); z.yang@pku.edu.cn (Z.Y.); haoyl@pku.edu.cn (Y.H.)

**Keywords:** small-sized, acoustic horn, thermal acoustic particle velocity sensor, sensitivity enhancement

## Abstract

In this paper, small-sized acoustic horns, the sensitivity enhancement package for the MEMS-based thermal acoustic particle velocity sensor, have been designed and optimized. Four kinds of acoustic horns, including tube horn, double cone horn, double paradox horn, and exponential horn, were analyzed through numerical calculation. Considering both the amplification factor and effective length of amplification zone, a small-sized double cone horn with middle tube is designed and further optimized. A three-wire thermal acoustic particle velocity sensor was fabricated and packaged in the 3D printed double cone tube (DCT) horn. Experiment results show that an amplification factor of 6.63 at 600 Hz and 6.93 at 1 kHz was achieved. A good 8-shape directivity pattern was also obtained for the optimized DCT horn with the lateral inhibition ratio of 50.3 dB. No additional noise was introduced, demonstrating the DCT horn’s potential in improving the sensitivity of acoustic particle velocity sensors.

## 1. Introduction

Sound signal is one of the most common forms of signals in nature. To fully define a sound signal at a single point, at least two parameters are required, a scalar of acoustic pressure and a vector of acoustic particle velocity [1]. Sound propagates through the reciprocal motion of medium particles, and the vibration velocity of medium particles is what we referred as “acoustic particle velocity”. The acoustic particles or medium particles are not the medium molecules that perform irregular thermal motions. They are small medium volumes containing many molecules so that the effects of thermal agitation can be expected to be averaged out.

The MEMS-based thermal acoustic particle velocity sensor (TAPVS) is a kind of sensor that can directly measure the particle velocity of sound signals [2,3,4,5,6]. In air, sound propagates in the form of longitudinal waves. Thus, the direction of sound propagation can be obtained by measuring the direction of acoustic particle velocity. TAPVS shows promising properties for applications in sound source localization [7,8,9,10,11,12,13], such as engine fault detection [14,15,16,17], gunshot localization [18,19], and pipe leakage acoustic detection [7,20] etc., which is far superior to microphone arrays for its higher orientation accuracy, smaller size, simpler signal processing [21], wider working temperature range [22], lower low frequency self-noise [4,23], and so on.

The core sound sensing unit of TAPVS is composed of three extremely fine thermistor microbeams, including one heating beam in the middle to form and maintain the self-built temperature field and two temperature sensing beams with extremely small heat capacity arranged parallel and symmetrically on both sides to convert the acoustic induced temperature disturbance to resistance difference. At the same time, these microbeams are very fragile and easily affected by the environment. However, in current research on TAPVS [20,24,25], sensors used in the experiments are all directly exposed to the measurement environment, without much additional protection. Although in this way the high sensitivity of the sensor can be well maintained, it also allows dust, moisture, and salt spray in the environment to directly contact the sensor chip, which may have a serious impact on the lifetime of the sensor. In addition, without any package protection, TAPVS’s applications in real scenarios are greatly limited, especially in harsh environments. In this case, an acoustic-transparent skin is badly needed to be used for a TAPVS protection package to reduce the environmental impact.

Another problem occurring while packaging TAPVS in a protection skin is that even if the sound-transmitting material is used, the transmission loss [26,27,28,29] of the sound signal is inevitable, which will seriously influence the sensitivity of TAPVS. Therefore, it is necessary to develop an effective protocol to maintain its high sensitivity while TAPVS is protected by sealing covering. The acoustic horns are usually used to amplify acoustic signals, which seem to be a good solution. Actually, the horns can improve the sensitivity of TAPVS significantly, no matter if the protection skin exists or not (Figure 1).

Acoustic horns were thoroughly analyzed by Webster in 1919 [30]. Based on the assumptions that the acoustic wave traveled only along the axial direction and that the sound energy was uniformly distributed over the cross-section in the direction of sound propagation, Webster simplified the three-dimensional wave equation into one-dimensional, and proposed the Webster’s horn equation. Later, Stuart derived the solutions of exponential, parabolic, and hyperbolic horns from Webster’s horn equation [31]. Donskoy et al. derived the formulation of double cone horn from Webster’s horn equation and verified the horn amplification factors using numerical modeling [32,33,34]. Honschoten et al. analyzed the double cylinder horn both analytically and by means of finite volume simulations on fluid dynamics [35].

Previous studies on acoustic horns based on Webster’s equation are all about large-sized horns, where the throat radius of the horn is far greater than the thickness of the viscous boundary layer caused by the viscosity of the medium and that the boundary layer effect can be ignored. The assumptions of the Webster horn equation are valid, and the previous conclusions are still available. As for MEMS-based TAPVS, acoustic horns should be scaled down to a smaller size, where the boundary layer effect is no longer neglectable and the assumptions of the Webster’s horn equation may also be invalid. To address this problem, a correction model for the small-sized double cone (DC) horn with the consideration of the boundary layer effect was proposed and experimentally verified in our previous work [36], which can give more accurate velocity gain amplification factors at the narrow horn throat than the traditional DC horn model [33]. Although such a model can give us insights to the horn design, it has limitations for exploring more complicated horn structures.

To further pursue the optimized horns suitable for MEMS-based TAPVS, a numerical modeling and optimization method is developed in this paper, which emphasizes the analysis of acoustic properties of different shapes of small-sized acoustic horns for MEMS-based TAPVS. In Section 2, numerical analysis is carried out to test the influence of the boundary layer effect on the acoustic horns with different sizes. Simulations on the acoustic properties of different kinds of horns, including velocity gain at horn throat, length of amplification zone, and directivity are carried out. Then, an optimized horn shape and horn size are given. In Section 3, a three-wire TAPVS was fabricated for verification of the above simulation results. And in Section 4, experiments are carried out and the results are discussed in detail.

## 2. Theory and Simulation

### 2.1. Theory

Acoustic waves are produced by vibration of particles in the medium. In a fluid medium such as air and water, solutions of acoustic problems can be found by solving hydrodynamic equations. Since sound waves are compressed waves under small acoustic disturbance, the variation of sound pressure, temperature, velocity, and density caused by the sound field can be treated as perturbation compared to the steady-state medium. Besides, for the case of no medium flow, the fluid velocity equals the particle vibration velocity, and the hydrodynamic equations [37] can be simplified as shown in Equation (1):(1)$$\rho_{0}\frac{\partial u_{t}}{\partial t}=\nabla \cdot \left[-p_{t}I+\mu \left(\nabla u_{t}+{\left(\nabla u_{t}\right)}^{⊤}\right)-\left(\frac{2}{3}\mu -\mu_{B}\right)\left(\nabla \cdot u_{t}\right)I\right]$$
where $\rho_{0}$ is the background density, $p_{t}$ and $u_{t}$ are fluctuations in pressure and velocity over the background values, respectively. Since the thermal conductivity of the medium and the temperature gradients of the disturbance are small enough that no appreciable thermal energy transfer occurs between adjacent medium elements, the adiabatic process can be assumed. Besides, when sound propagates in the free field, the influence of the medium viscosity is small and can be ignored, the equation above can be further simplified into the sound pressure wave equation [38], as shown in Equation (2):(2)$$\nabla \cdot \left(-\frac{1}{\rho_{0}}\nabla p\right)-\frac{1}{\rho_{0}}{\left(\frac{\omega }{c}\right)}^{2}p=0$$
where p is the sound pressure and the only parameter to be solved in the frequency domain. The particle velocity is calculated from Equation (3) [39]:(3)$$\rho_{0}\frac{\partial }{\partial t}u\left(x,t\right)+\frac{\partial }{\partial x}p\left(x,t\right)=0$$

Numerical calculations based on Equations (1) and (2) respectively are used to calculate the acoustic particle velocity amplification factors $G_{A}$ of two double cone (DC) acoustic horns with different sizes, where $G_{A}$ is expressed as: the ratio of the particle velocity at the horn throat ($v_{t}$) and particle velocity at the same position but without acoustic horns ($v_{0}$):(4)$$G_{A}=\frac{v_{t}}{v_{0}}$$

The numerical calculation results are shown in Figure 2. The parameters of the horns are shown in Table 1. Figure 2a describes the $G_{A}$ of the large-sized DC horn with and without consideration of the boundary effect in the frequency range from 10 to 10 kHz. It can be obviously seen that there is little difference of the two $G_{A}$ frequency response curves in the low frequency range for the large-sized horn and that the viscosity is neglectable. However, for the small-sized horn, as shown in Figure 2b, greater particle velocity gains are observed especially in the low frequency range when the medium viscosity is taken into consideration. The reason is that the size of the horn and the thickness of viscous boundary layer are in comparable scale, where the boundary layer effect has a significant influence on the results.

### 2.2. Simulation

Numerical simulations are further performed by using the finite element software COMSOL Multiphysics to simulate the characteristics of different types of small-sized acoustic particle velocity horns.

According to the relationship between horn cross-section area S(x) and x, four kinds of horn including tube (S(x)~const), double cone (DC) horn (S(x)~x^2^), double quadratic parabola (DP) horn (S(x)~x^4^), and double exponential (DE) horn (S(x)~e^mx^) are simulated.

Since there are the viscous boundary layer and thermal boundary layer near the wall, and the viscous loss gradient caused by shear and heat conduction is large, the thermo-viscous acoustic module is used for simulation. Because of their rotational symmetry, the axial sections of the four kinds of acoustic velocity horns are selected as the calculation domain, as shown in Figure 3. The R_T_, R_M_, and L of the horns are 0.5, 3, and 3 mm, respectively. For the tube horn, the diameter of the horn throat equals the diameter of the horn mouth. Besides, since the acoustic impedance of the air medium is much smaller than that of the solid package, the wall of the acoustic velocity horn is set as the sound hard boundary. The boundary of the air domain is set as the impedance boundary condition and the adiabatic pressure is used at the left side of the medium boundary for the purpose of the acoustic source. In addition, the thickness of the viscous boundary layer is meshing separately. The cross-sectional distribution of acoustic particle velocity at 500 Hz for different horns is shown in Figure 3. An obvious velocity amplification can be seen at the throat of the acoustic horns except for the tube horn, but the velocity amplification area is restricted in a narrow zone around the center of the horn throat, as shown in Figure 3.

As for TAPVS, the acoustic sensitive unit is usually hundreds of microns wide. The particle velocity distribution along the *x*-axis, the most sensitive direction of TAPVS packaged in the horns, is then simulated and the results are shown in Figure 4. The maximum velocity, vmax is observed at the DP horn throat, which is slightly larger than that of the DC horn and DE horn. The velocity in the tube horn is not amplified very much but its velocity distribution is quite flat. As for the other three kinds of horns, the acoustic particle velocity decreases rapidly with the increasing distance from the horn throat. The inset graph shows the velocity distribution around the horn throat.

Besides, the length of the amplification zone with an acoustic particle velocity greater than 99% of v_max_, 95% of v_max_, and 90% of v_max_, respectively, are calculated and the results are shown in Figure 5a. The length of the amplification zone indicates the effective assembly range around the horn throat within a certain fluctuation of acoustic particle velocity. For the DC horn, only packaged within 67 μm around the center of the horn throat, the velocity fluctuation as well as the systematic error is less than 1%. The larger the size of the acoustic sensitive unit, the larger the acoustic particle velocity fluctuation around the horn throat, and the larger the systematic error. Different kinds of horns have different effective amplification zone length. The DE horn is better than the DC horn but worse than the DP horn. The tube horn is much better than the other three kinds of horns for its rather flat velocity distribution, but on the other hand, much worse for its little velocity amplification effect. There is a trade-off between the amplification factor and the length of the amplification zone. The decision can be made by replacing the throat of the other three kinds of horns with the tube horn, as shown in Figure 5b.

In order to characterize the influence of the small-sized middle tube on the throat velocity amplification factor $G_{A}$ of the whole acoustic horns, simulations are performed and the frequency responses of the amplification factor for the DCT horn, DPT horn, and DET horn, respectively, are shown in Figure 6a. In contrast, the results of the DC, DP, and DE horn are also depicted. Three conclusions can be drawn in the figure. First, the $G_{A}$ curves decrease in the same way with the increase of sound frequency, which is in accordance with previous simulation results in Figure 2. Second, the $G_{A}$ curve of the DPT horn is the largest, which is slightly greater than that of DCT and DET. The $G_{A}$ curves of these horns show little difference at the same sound frequency, where the $G_{A}$ of the DPT horn is only 1.27% greater than that of the DCT horn at 500 Hz and 1.79% greater at 1000 Hz. Third, $G_{A}$ of acoustic horns with a small-sized middle tube is greater than that without a middle tube. As shown in Figure 6b,c, the velocity boundary of DCT is larger than that of DC, which makes the effective radius of horn throat of DCT smaller than DC. The larger the ratio of the effective radius of the horn mouth and radius of the horn throat, the larger the $G_{A}$. Thus, for small-sized acoustic horns, horns with a middle tube are better than those without. Considering the differences of DCT, DPT, and DET are small while machining of the DCT horn is much easier than that of the DPT and DET horn, the DCT horn is preferable. 

The influence of all the horns with and without the middle tube on the directivity of TAPVS is simulated by applying a monopole point sound source with different incident directions. The results are shown in Figure 7. All the horns show good 8-shape patterns, indicating little influence on the directivity of TAPVS.

To summarize, the DCT horn is the best among the acoustic horns discussed above. Further studies were carried out on the optimization of the size of the DCT horn designed for TAPVS.

The thickness of MEMS-based TAPVS is about 250 μm, and the velocity boundary layer thickness $\delta_{v}$ is smaller than 200 μm when the frequency range is larger than 100 Hz, which is given by [39]:(5)$$\delta_{v}=\sqrt{2\frac{\mu }{\omega \rho_{0}}}$$
where *μ* is the dynamic viscosity of the medium and *ω* is the angular frequency of the sound signal. Therefore, 0.5 mm is necessary and enough for the design of R_T_. As for R_M_ and L, simulations are performed to find the optimal values. The simulation results are shown in Figure 8. As can be seen from the figure, the amplification factor first quickly increases with the increase of R_M_ and then quickly decreases, and the optimal R_M_ decreases with the increase of L and sound frequency. Therefore, the optimal R_M_ value is related to the value of L and the sound frequency. The amplification factor changes slowly with the increase of L, thus the impact is small for the L value. Considering that the large size of package may lead to the distortion of the sound field, the value of L should be as small as possible. The final optimal designed values of R_M_ and L for the 1.6 mm-diameter horn throat DCT are chosen for 3 and 8 mm, respectively, and the simulated amplification factor is 5.6 at 500 Hz and 4.0 at 2000 Hz. As for the length of the middle tube of DCT, it should be as short as possible in the case of meeting the assembly tolerance. Besides, although the thickness of the horn package has no influence on the velocity amplification factor, it should be as small as possible because the larger package is more likely to cause the distortion of the sound field. Process deviations of Rt, Rm, and L should also be as small as possible because they are closely related to the throat velocity amplification factor of the acoustic horns.

## 3. Fabrication

In order to experimentally verify the amplification factor of DCT, a three-wire MEMS-based thermal acoustic particle velocity sensor was fabricated. Figure 9 shows the key steps of the fabrication process.

Firstly, thermal oxidation of 300 nm silicon oxide (SiO_2_) was performed on silicon substrate as the stress buffer layer. After that, the 100 nm silicon nitride (Si_3_N_4_) layer and another 20 nm silicon oxide layer were deposited as the supporting layer by low pressure chemical vapor deposition (LPCVD), as shown in Figure 9a. Then, the first photolithography was performed. Photoresist was spin-coated and patterned. The exposed SiO_2_/Si_3_N_4_/SiO_2_ layer was then etched by reactive ion etching (RIE). After that, the exposed silicon was isotropic etched, which undercuts part of the silicon under the beams, as shown in Figure 9b. Then, a second photolithography was performed to pattern electrodes and thermistors. Physical vapor deposition (PVD) was conducted to form 10 nm chromium (Cr) as an adhesion layer and 200 nm platinum (Pt) as a thermosensitive layer was followed. Right after that, the lift-off process was performed, as shown in Figure 9c. Finally, RIE was used to remove the backside SiO_2_/Si_3_N_4_/SiO_2_ layers. All the beams were released by tetramethyl-ammonium hydroxide (TMAH) anisotropic etching, as shown in Figure 9d–f of the SEM photographs of an TAPVS chip and part of the thermistor beam.

The horns were fabricated by 3D printing. The material used is photosensitive resin with a density of 1.15 g/cm^3^ and sound speed of 2540 m/s. Its characteristic acoustic impedance is 7302 times that of air. Thus, the horns can be regarded as the sound hard boundary while testing in air. Other materials with large density and large sound velocity like copper, stainless steel, aluminum alloy, etc., can also be used and function normally, but need a special package design to avoid the short circuit of TAPVS. Besides, the machining difficulty of the small-sized horn package is much greater than 3D printing. While for longer term use, the organic material is more likely to suffer abrasion and contamination. Thus, for those applications in harsh environments, an air-tight sealing covering should be packaged outside the horn packaged TAPVS, and the horn is protected at the same time. Thus, the 3D-printed horn can stand long-term use in practical application domains.

For comparison, both the DCT horn and the DC horn were fabricated with the same size, where Rt, Rm, and L of the DC horns are designed to be 0.5, 3, and 3 mm, respectively. The length of the tube in the DCT horn is 1 mm. A little bit larger but with optimized acoustic properties, the DCT horn was also fabricated, with the optimized Rt, Rm, and L to be 0.5, 8, and 3 mm, respectively. The schematic of the DC horn, DCT horns, and TAPVS packaged in a DC horn, and the photo of the packaged TAPVS in the DCT horn and optimized DCT horn are shown in Figure 10. A misaligned DC sensor was also prepared, as shown in Figure 10g, where the acoustic sensitive center of the TAPVS is misaligned with the center of the DC horn throat. As shown in Figure 6c, the velocity difference is small, within 0.6 mm around the center of horn throat along the *y*-axis, and the thickness of TAPVS is only 0.25 mm, thus the sensor attachment deviations along the *y*-axis can be ignored. While along the *x*-axis, a sensor assembly groove was made to make sure the sensor was attached in a fixed position, as shown in Figure 10a.

## 4. Experiments

### 4.1. Experimental Setup

The amplification factor test of DC and DCT horns was carried out in a standing wave tube (SWT), as shown in Figure 11. The cross-section of the SWT is circular with the diameter d_SWT_ of 10 cm. The cut-off frequency of the SWT is 1988 Hz, which is given by [23]:(6)$$f_{cut-off}=\frac{c}{1.71d_{SWT}}$$
where *c* is the sound velocity of the air. The packaged TAPVS is fixed on the rotation axis of a calibrated rotary stage, aligning the 1/2-inch test hole 10 cm away from the SWT bottom reflector. A standard microphone is fixed on the SWT on the same cross section as the test hole. When the sound frequency is below the SWT cut-off frequency, the acoustic impedance $Z_{a}$ in SWT is given by [2]:(7)$$Z_{a}=p\left(x\right)/v\left(x\right)=-i\rho c\cdot \cot\left(k\left(l-x\right)\right)$$
where *k* is the wavenumber of sound signals, *l* is the length of the SWT, and *x* is the distance between the packaged sensor and the speaker. As long as the acoustic pressure is measured, the exact particle velocity can be calculated from expression (7), and the sensitivity of the sensor can be measured.

### 4.2. Experimental Results and Discussion

Firstly, responses of TAPVS with and without acoustic horns (marked as DCT sensor, DC sensor, misaligned DC sensor, optimized DCT sensor, and bare sensor, respectively) to particle velocity at both 600 and 1000 Hz were measured and the results are shown in Figure 12. The results suggest that the three kinds of packaged sensors all respond linearly with acoustic particle velocity. In addition, with the same acoustic particle velocity, responses of the TAPVS packaged in the DCT horn are smaller than that in the DC horn but greater than the bare sensor, which is in good accordance with the previous analysis.

Frequency responses of the five kinds of sensors were measured in the frequency range of 200 to 1500 Hz, and the results are shown in Figure 13. The amplification factor in Figure 13b is obtained through the ratio of the sensitivity of the horn-packaged sensor and the bare sensor. As can be seen from Figure 13a, the optimized DCT sensor shows the largest sensitivity. The sensitivity of the DCT sensor is slightly larger than that of the DC sensor, which agrees well with the simulation results in Figure 6a. The sensitivity of the misaligned DC sensor is smaller than that of the DC sensor, indicating that the assembly error does have a bad influence on TAPVS sensitivity. All the horn-packaged sensors have larger sensitivity than that of the bare sensor, showing the effectiveness of the acoustic horns.

Both the measured amplification factor and the simulated results are shown in Figure 13b. The misaligned DC sensor shows the smallest amplification factor. The measured amplification factor curves of DC (in red circles) and DCT (in blue triangles) agree well with the simulation results (purple short dash and blue short dash, respectively). The measured amplification factors of the optimized DCT sensor are smaller than the simulation results in the frequency range of 200–800 Hz. The possible reason is the sound leakage at the test hole on the wall of the standing wave tube. The relatively large size of the optimized DCT horn makes it more difficult to realize good occlusion. The influence of the sound leakage increases with the decrease of the sound frequency. Besides, the optimized DCT horn has a relatively large influence on the standing wave field near the wall of the 10 cm-diameter standing wave tube than the bare sensor for its relatively large package size. Thus, the ratio of the measured results of the optimized DCT sensor and the bare sensor is also influenced. In spite of the measurement error, the velocity amplification factor of the optimized DCT is greater than that of DCT. Compared with the DCT sensor, the velocity amplification factor of the optimized DCT is 6.63 at 600 Hz and 6.93 at 1 kHz, improved by 85% and 99%, respectively.

The directivity at both 600 and 1000 Hz of the five kinds of sensors were measured in the SWT with the help of a rotary stage as shown in Figure 11, and the results are shown in Figure 14. All the sensors show good 8-shape directivity patterns. The lateral inhibition ratio, the ratio of the maximum sensitivity and the minimum sensitivity, is 44.8 dB for misaligned DC sensor, 50.6 dB for DC sensor, 50.8 dB for DCT sensor, and 50.3 dB for optimized DCT sensor, showing their excellent directivity. Considering the initial angle of manual installation of TAPVS on the rotary stage, there might be an angle of 37 ± 5° between the most sensitive direction of TAPVS and the zero-tick mark. The directivity of TAPVS with acoustic horns can be further improved by fine tuning the installation angle of the sensor.

Furthermore, noise of the TAPVS with different packages was measured, and the results show little difference, as depicted in Figure 15. The total measurement noise can be divided into two parts: thermistor thermal noise and circuit noise. Since all of the tests are carried out under the same conditions but different sensor packages, the circuit noise is at the same level. Besides, tested in the same working medium, the temperature of the sensing beams should also be the same.

## 5. Conclusions

In this paper, small-sized acoustic horns were designed and optimized for thermal acoustic velocity sensors taking into account of the boundary layer effect. Comparisons of acoustic horns with different dimensional sizes were given through numerical analysis. Four kinds of acoustic horns with different cross-sections were simulated by finite element simulation. Considering both the amplification factor and the amplification zone length, the DCT horn is chosen and optimized through the values of L and Rm. The final optimal designed values of Rm and L for the 1 mm-diameter horn throat DCT are 8 and 3 mm, respectively. 

A three-wire thermal acoustic particle velocity sensor was fabricated to experimentally verify the amplification factor of DCT. 

Both the DCT horn and the DC horn were fabricated with the same size, where Rt, Rm, and L of the DC horns are designed to be 0.5, 3, and 3 mm, respectively. The length of the tube in the DCT horn is 1 mm. A little bit larger but with optimized acoustic properties, the DCT horn was also fabricated, with the optimized Rt, Rm, and L to be 0.5, 8, and 3 mm, respectively. A misaligned DC sensor was prepared for comparison.

The experiments were carried out in the standing wave tube. A good linear relationship was observed through the response curve of TAPVS to acoustic particle velocity. 

Frequency responses of the bare sensor, misaligned DC sensor, DC sensor, DCT sensor, and optimized DCT sensor were measured in the frequency range of 200–1500 Hz. The optimized DCT sensor shows the largest sensitivity. The sensitivity of the DCT sensor is slightly larger than that of the DC sensor, which agrees well with the simulation results. The sensitivity of the misaligned DC sensor is smaller than that of the DC sensor. Further, the amplification factors of the five kinds of sensors were calculated through the ratio of the sensitivity of the packaged sensor and the bare sensor. For the optimized DCT sensor, the velocity amplification factors are 6.63 at 600 Hz and 6.93 at 1 kHz, improved by 85% and 99%, respectively, compared with the DCT sensor.

The directivity at both 600 and 1000 Hz of the three kinds of packaged sensors were measured. The lateral inhibition ratio is 44.8 dB for misaligned DC sensor, 50.6 dB for DC sensor, 50.8 dB for DCT sensor, and 50.3 dB for optimized DCT sensor, showing their excellent directivity. 

Noise of the TAPVS with different packages showed little difference, indicating that not only the sensitivity but also the signal to noise ratio are improved through acoustic horns.

## Figures and Tables

**Figure 1 sensors-21-04337-f001:**
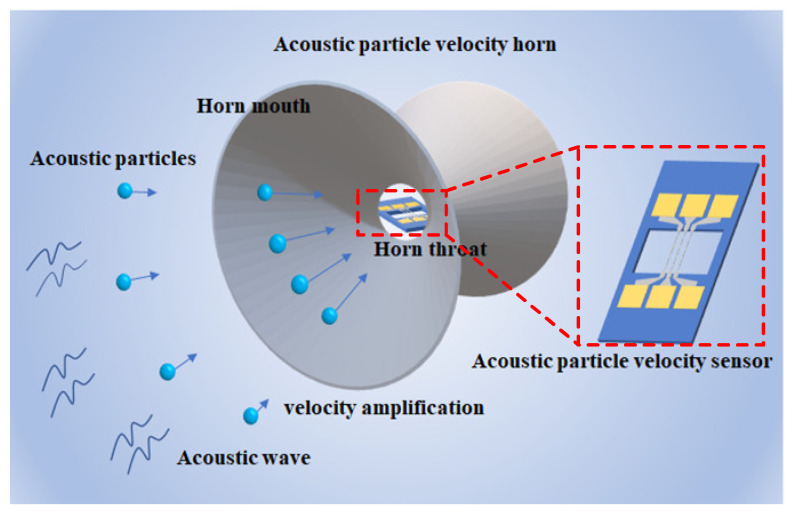
The diagram of a TAPVS packaged in an acoustic horn.

**Figure 2 sensors-21-04337-f002:**
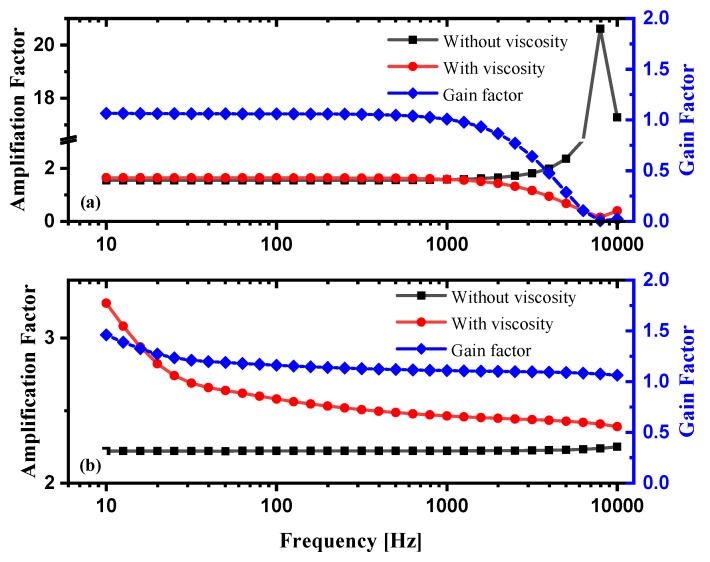
Frequency response curves of: (**a**) amplification factor for Horn 2, the large-sized DC horn with and without consideration of the viscosity of the medium, and the ratio of the two amplification factors; (**b**) amplification factor for Horn 1, the small-sized DC horn with and without consideration of the viscosity of the medium, and the ratio of the two amplification factors.

**Figure 3 sensors-21-04337-f003:**
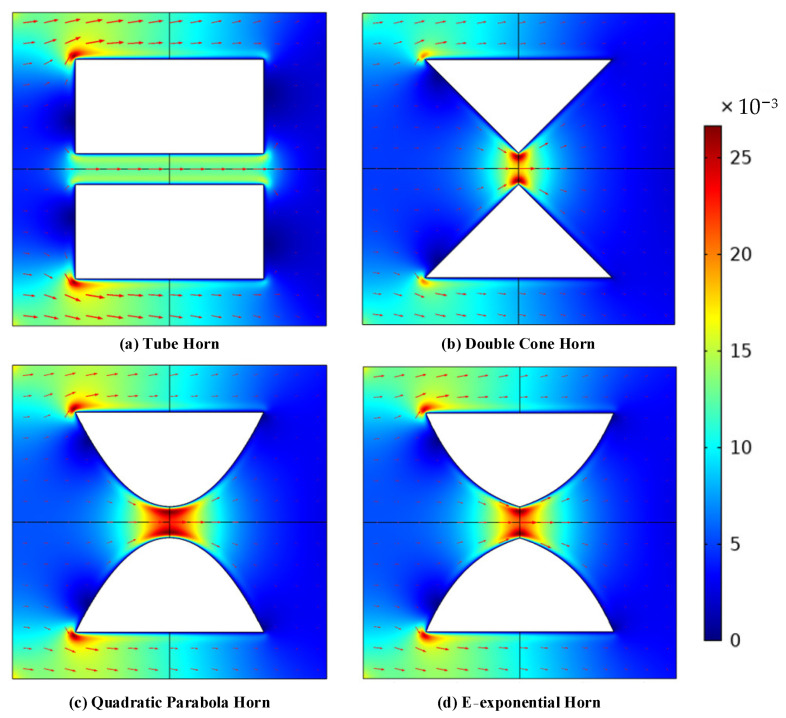
Acoustic particle velocity cross-sectional distribution at 500 Hz for: (**a**) tube horn; (**b**) DC horn; (**c**) DP horn; (**d**) DE horn.

**Figure 4 sensors-21-04337-f004:**
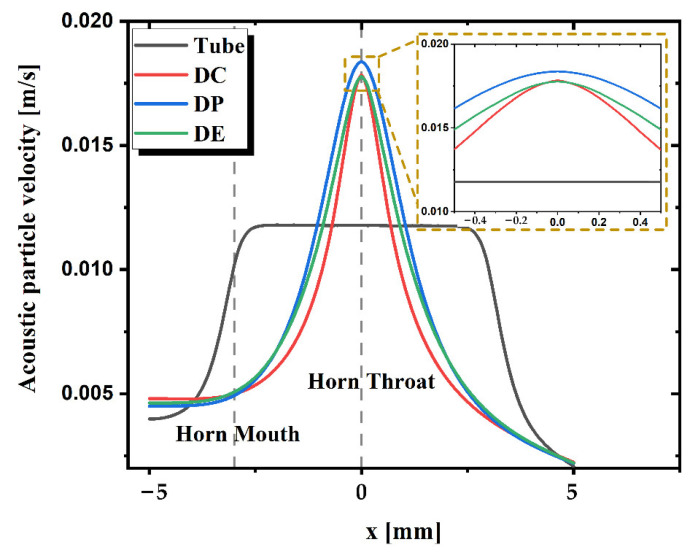
Acoustic particle velocity distribution along the *x*-axis of the tube horn, DC horn, DP horn, and DE horn. The inset shows the velocity around the horn throat.

**Figure 5 sensors-21-04337-f005:**
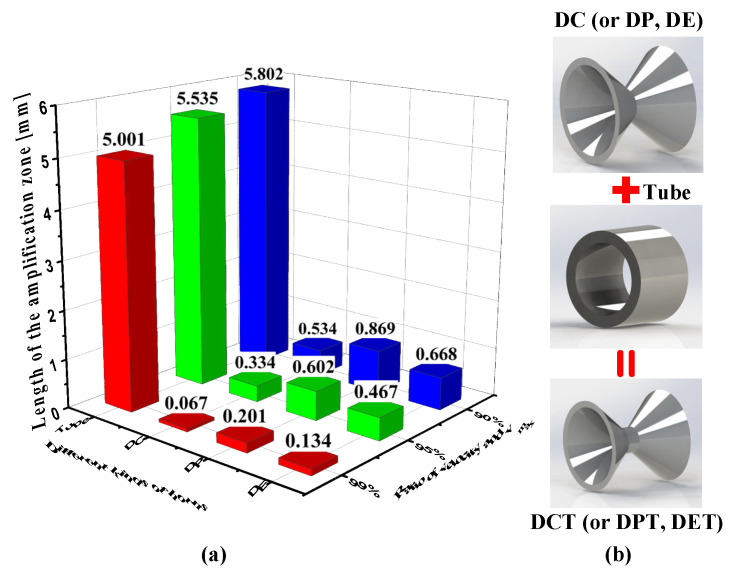
(**a**) The length of amplification zone of four different kinds of acoustic horns with acoustic particle velocity greater than 99%, 95%, and 90% of v_max_, respectively. (**b**) Replacing the throat of DC horns with the tube horn.

**Figure 6 sensors-21-04337-f006:**
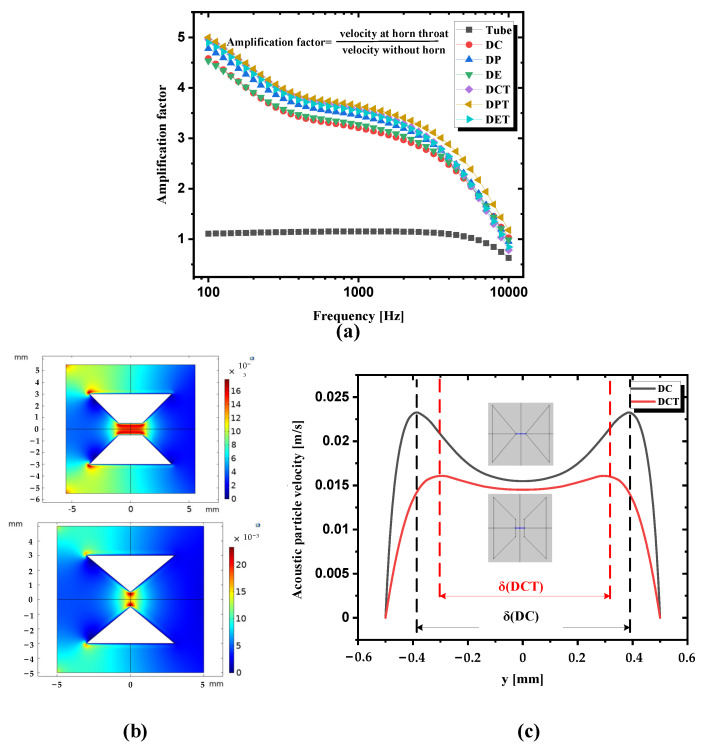
(**a**) Frequency responses of the velocity amplification factor for different kinds of acoustic horns. The amplification factor is defined by the ratio of acoustic particle velocity at the horn throat and velocity without the horn; (**b**) Acoustic particle velocity distribution of the DCT horn and DC horn; (**c**) Velocity distribution along the *y*-axis at the DCT horn throat and DC horn throat.

**Figure 7 sensors-21-04337-f007:**
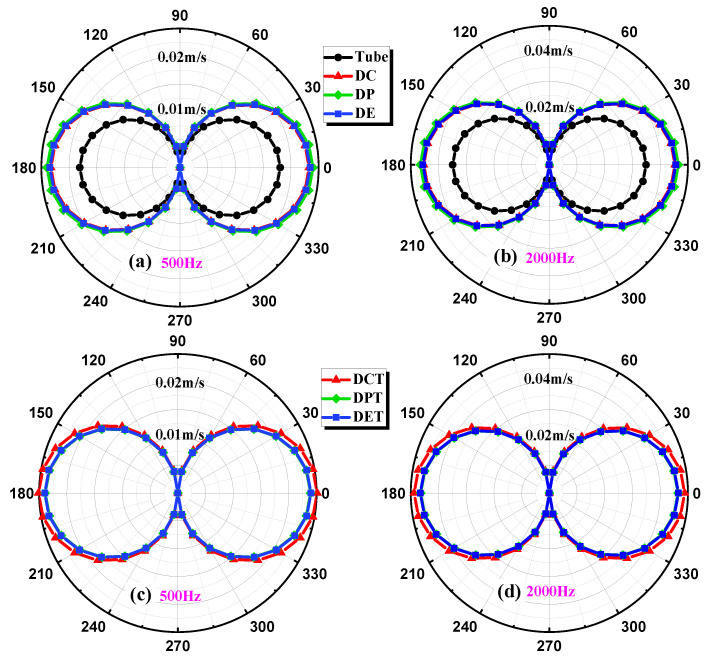
Directivity patterns of different kinds of acoustic horns: (**a**) without middle tube at 500 Hz; (**b**) without middle tube at 2000 Hz; (**c**) with middle tube at 500 Hz; (**d**) with middle tube at 2000 Hz.

**Figure 8 sensors-21-04337-f008:**
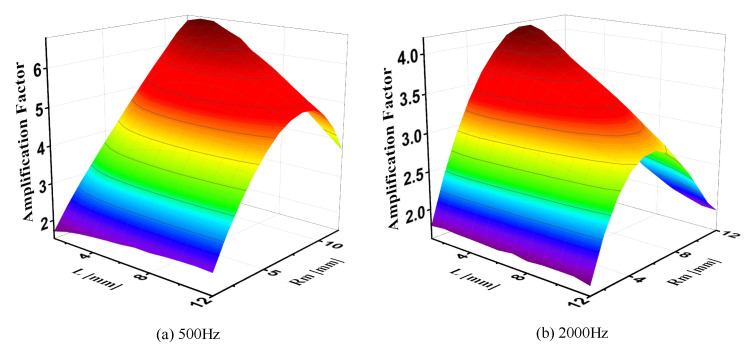
Amplification factor of the DCT horn with different L and Rm at: (**a**) 500 Hz and (**b**) 2000 Hz.

**Figure 9 sensors-21-04337-f009:**
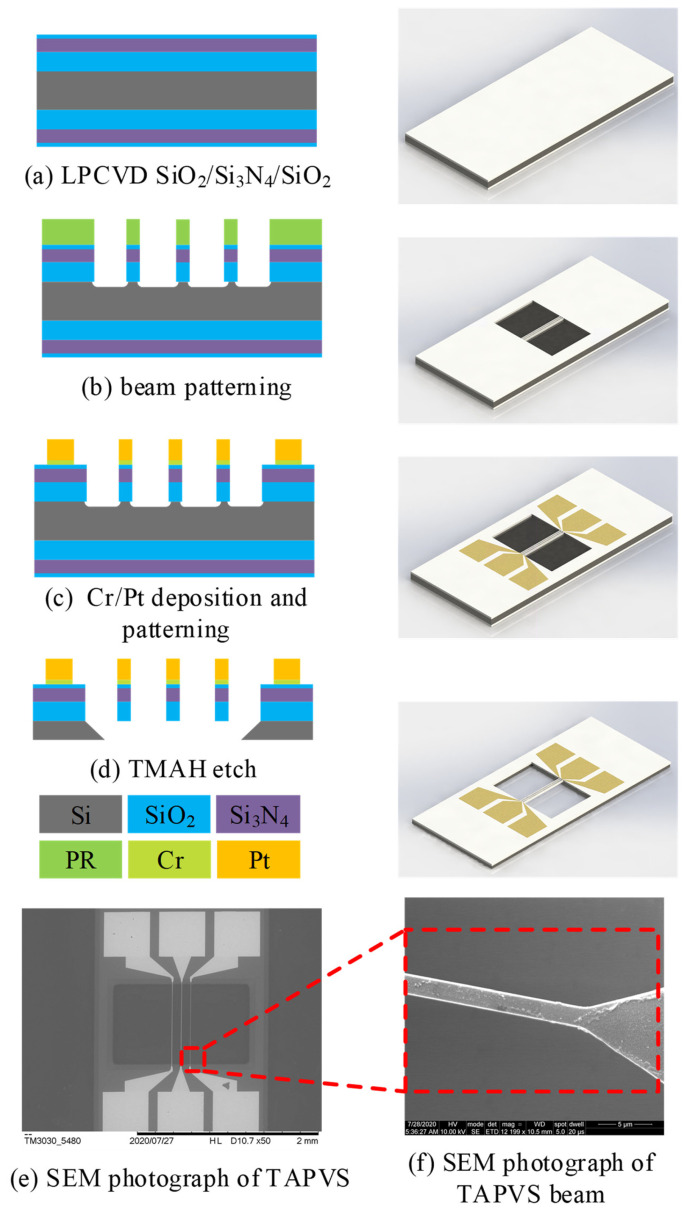
Key steps of the fabrication process of TAPVS: (**a**) deposition of SiO_2_/Si_3_N_4_/SiO_2_; (**b**) patterning the supporting layer of the beams and RIE, the exposed SiO_2_/Si_3_N_4_/SiO_2_, after the first photolithography; (**c**) deposition and lift-off of Cr/Pt; (**d**) releasing beams by TMAH anisotropic etching; (**e**) SEM photo of TAPVS; and (**f**) part of the thermistor beam.

**Figure 10 sensors-21-04337-f010:**
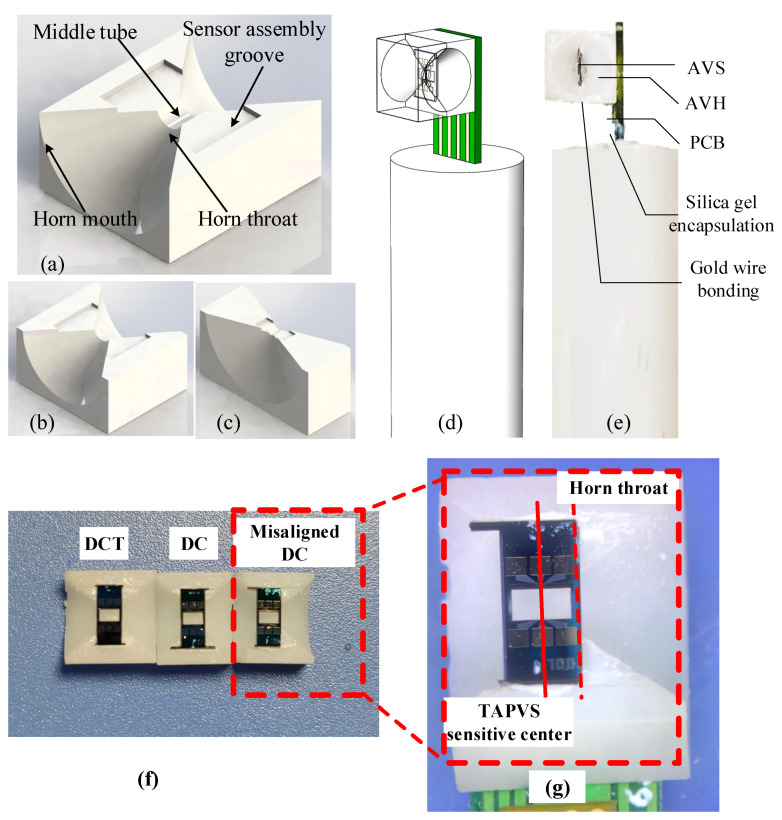
Schematic of: (**a**) DCT horn, (**b**) DC horn, (**c**) optimized DCT horn, (**d**) TAPVS packaged in a DC horn, and photo of (**e**) a TAPVS packaged in a DC horn; (**f**) TAPVS in different kinds of acoustics horns; (**g**) the misaligned DC sensor.

**Figure 11 sensors-21-04337-f011:**
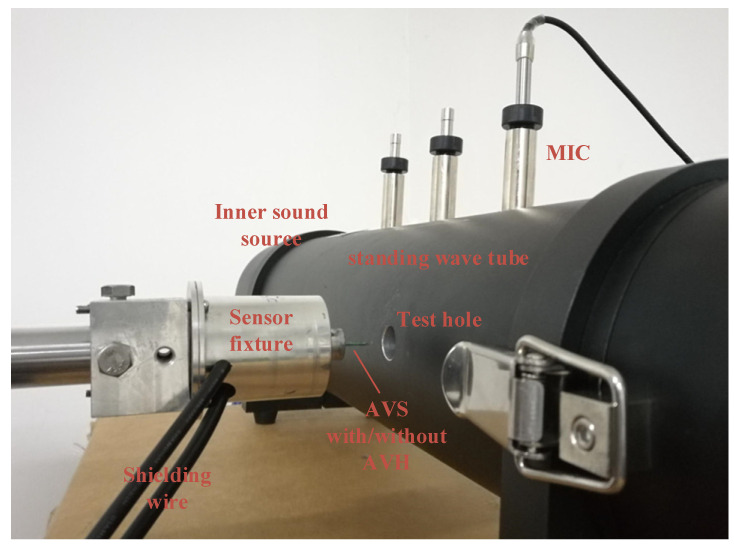
Photograph of the SWT used for the acoustic horns amplification factor test.

**Figure 12 sensors-21-04337-f012:**
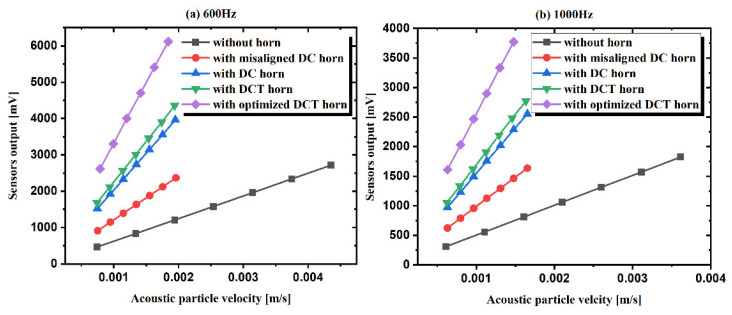
Responses of TAPVS with and without acoustic horns to particle velocity at: (**a**) 600 Hz and (**b**) 1000 Hz.

**Figure 13 sensors-21-04337-f013:**
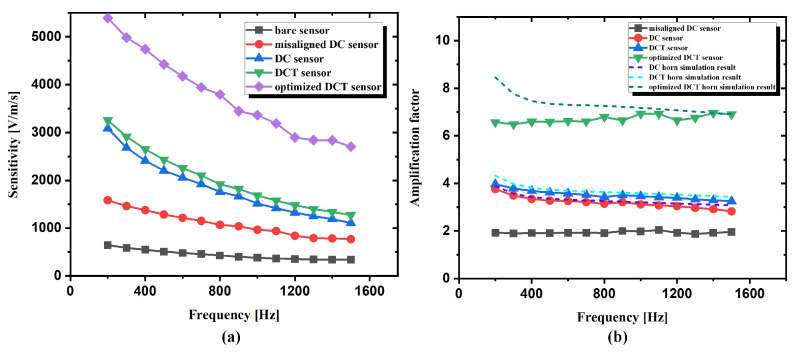
Frequency responses of: (**a**) TAPVS sensitivity with and without acoustic horns with the same designed parameters and (**b**) comparisons of measured and simulated amplification factors of TAPVS with and without acoustic horns.

**Figure 14 sensors-21-04337-f014:**
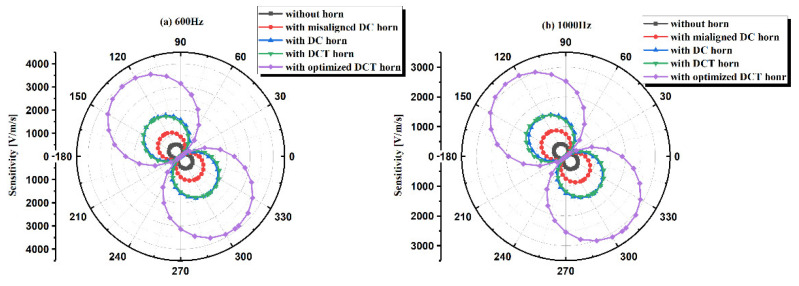
Directivity of TAPVS with and without acoustic horns at: (**a**) 600Hz and (**b**) 1000 Hz.

**Figure 15 sensors-21-04337-f015:**
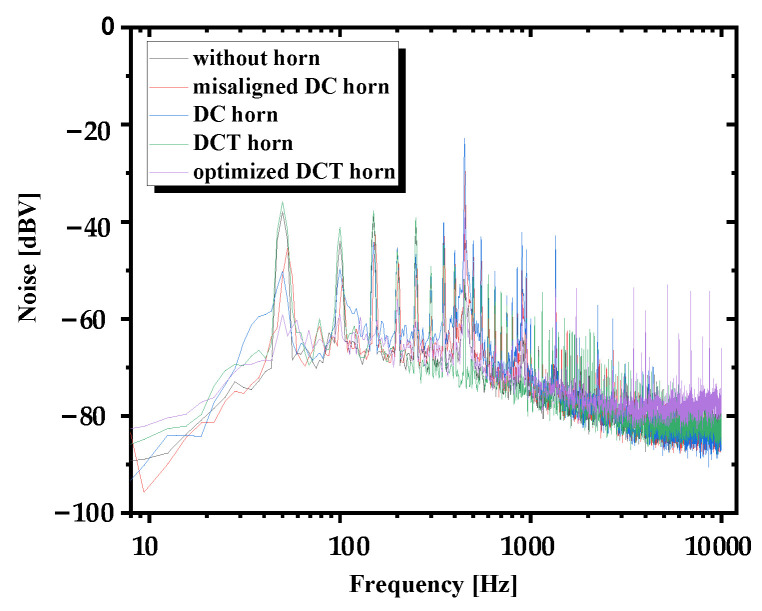
Noise of TAPVS with and without acoustic horns.

**Table 1 sensors-21-04337-t001:** Parameters of the acoustic horns with different sizes used in Figure 2.

	Radius of Horn Throat R_T_	Radius of Horn Mouth R_M_	Length of Horn L
Horn 1	0.5 mm	2 mm	5 mm
Horn 2	10 mm	30 mm	50 mm

## Data Availability

The data presented in this study are available on request from the corresponding author. The data are not publicly available due to restrictions of undergoing projects.

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
