# Peer review of "Design and Optimization of Sensitivity Enhancement Package for MEMS-Based Thermal Acoustic Particle Velocity Sensorâ€"

_sensors, 2021, doi:10.3390/s21134337_

Round 1
Reviewer 1 Report
The paper describes an interesting idea, but in some issues it seems not be self-contained/complete.
For instance, the notion of 'acoustic particle' should be more lucidly be explained, in particular, in the context of the regarded sound bearing medium. Also, why viscosity can be neglected and adiabatic process can be assumed, as well as sound-to-temperature transduction should be elucidated in more detail.
The discussion of the horn design is interesting, but only in one place a thickness (manufacturing material, I suppose) is mentioned.
But what is the effect of choice of horn material, thickness, and process deviations etc. on the regarded optimization and functionality.
What is the effect of sensor device to horn attachment ?
What about (expected) effects on longer term use (wear-out) in one of the practical application domains ?
Reviewer 2 Report
The authors presented the design and optimization of sensitivity improvement for MEMS velocity sensor. The design is effective which is virified by the experiment. However, in my opinion, a comparison between the presneted results and other devices should be added in the manuscript.
